# Par protein localization during the early development of *Mnemiopsis leidyi* suggests different modes of epithelial organization in the metazoa

**Miguel Salinas-Saavedra†\*, Mark Q Martindale\***

The Whitney Laboratory for Marine Bioscience, and the Department of Biology, University of Florida, St. Augustine, United States

**Abstract** In bilaterians and cnidarians, epithelial cell-polarity is regulated by the interactions between Par proteins, Wnt/PCP signaling pathway, and cell-cell adhesion. Par proteins are highly conserved across Metazoa, including ctenophores. But strikingly, ctenophore genomes lack components of the Wnt/PCP pathway and cell-cell adhesion complexes raising the question if ctenophore cells are polarized by mechanisms involving Par proteins. Here, by using immunohistochemistry and live-cell imaging of specific mRNAs, we describe for the first time the subcellular localization of selected Par proteins in blastomeres and epithelial cells during the embryogenesis of the ctenophore *Mnemiopsis leidyi*. We show that these proteins distribute differently compared to what has been described for other animals, even though they segregate in a host-specific fashion when expressed in cnidarian embryos. This differential localization might be related to the emergence of different junctional complexes during metazoan evolution.

**\*For correspondence:**
miguel.salinas-saavedra@
nuigalway.ie (MS-S);
mqmartin@whitney.ufl.edu (MQM)

**Present address:** †Centre for
Chromosome Biology,
Bioscience Building, National
University of Ireland Galway,
Galway, Ireland

**Competing interests:** The
authors declare that no
competing interests exist.

**Reviewing editor:** Patricia J
Wittkopp, University of
Michigan, United States

## Introduction

In bilaterians and cnidarians, a polarized epithelium is classically defined as a group of polarized cells joined by belt-like cell-cell junctions and supported by a basement membrane (*Magie and Martindale, 2008*; *St Johnston and Sanson, 2011*; *Thompson, 2013*; *Ohno et al., 2015*; *Salinas-Saavedra et al., 2015*). While the asymmetric cortical distribution of the Wnt Planar Cell Polarity (PCP) pathway components polarizes the cells along the tissue plane, the asymmetric cortical distribution of Par system components polarizes the cells along the apical-basal axis (*St Johnston and Sanson, 2011*; *Thompson, 2013*; *Gumbiner and Kim, 2014*; *Besson et al., 2015*; *Yang and Mlodzik, 2015*; *Ahmed and Macara, 2016*; *Aigouy and Le Bivic, 2016*; *Butler and Wallingford, 2017*; *Davey and Moens, 2017*; *Salinas-Saavedra et al., 2015*; *Fanto and McNeill, 2004*; *St Johnston and Ahringer, 2010*; *Cha et al., 2011*; *Kumburegama et al., 2011*; *Nance and Zallen, 2011*; *Momose et al., 2012*; *Wallingford, 2012*). The mechanisms that organize cell-polarity are highly conserved in all animals that have been studied and most likely been present in the most recent common ancestor (MRCA) of Cnidaria and Bilateria (*Thompson, 2013*; *Salinas-Saavedra et al., 2015*; *Kumburegama et al., 2011*; *Momose et al., 2012*; *Fahey and Degnan, 2010*; *Ragkousi et al., 2017*; *Salinas-Saavedra et al., 2018*; *Belahbib et al., 2018*; *Figure 1A*).

  Interestingly, ctenophores or comb jellies, whose position at the base of metazoan tree is still under debate (*Dunn et al., 2008*; *Hejnol et al., 2009*; *Ryan et al., 2013*; *Moroz et al., 2014*; *Whelan et al., 2017*), (*Simion et al., 2017*), (*Feuda et al., 2017*), possess a stereotyped development (*Figure 1B*) and do not have the genes that encode the components of the Wnt/PCP pathway in their genomes (*Ryan et al., 2013*). Thus, the study of the subcellular organization of the Par

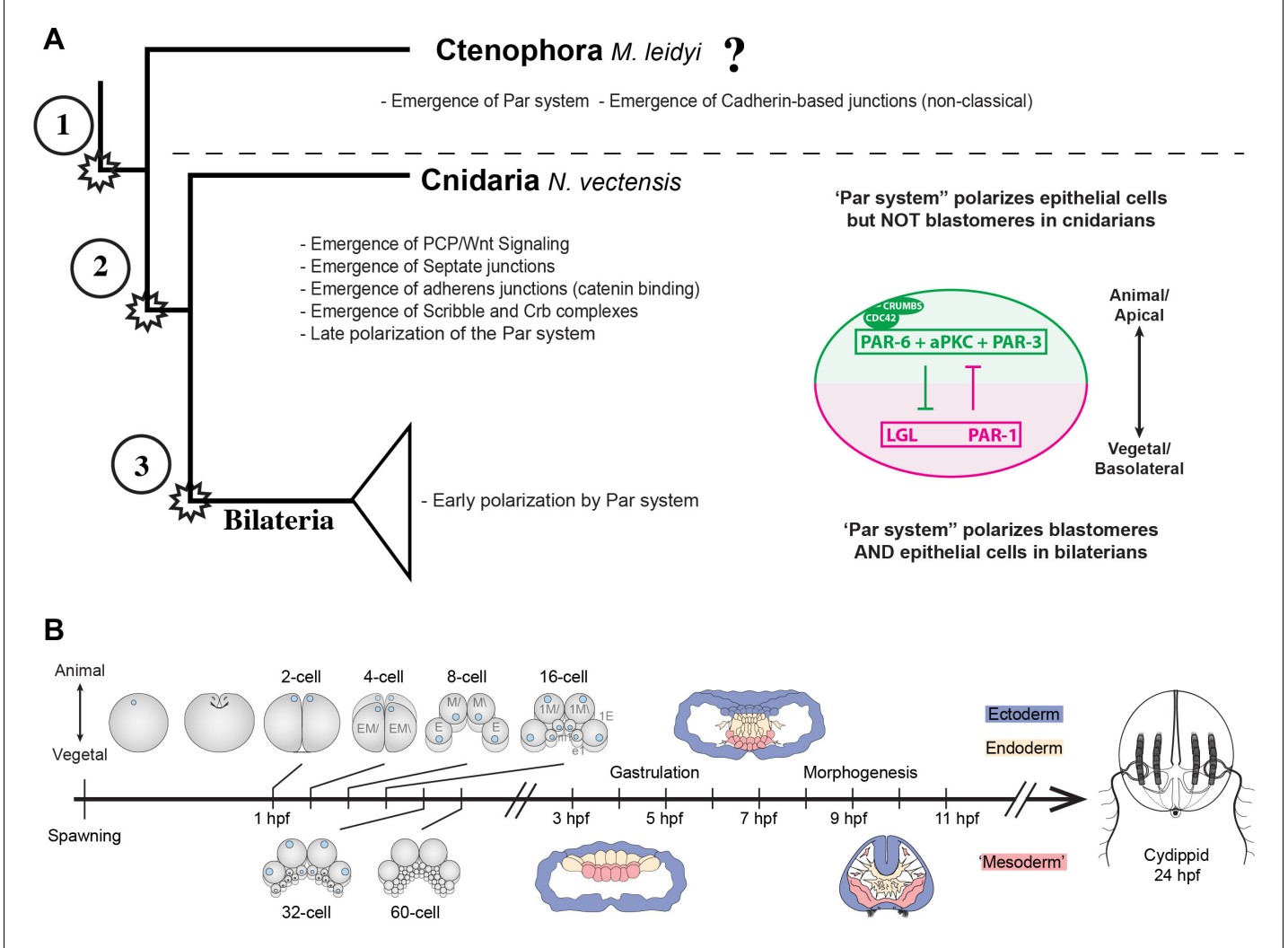

**Figure 1.** Evolution of cell polarity components during animal evolution. (**A**) Three major evolutionary steps (left side) that might have changed the organization of cell polarity in the Metazoa. The diagram (right side) depicts the subcellular asymmetric localization of Par proteins in Cnidaria and Bilateria. However, there are no previous descriptions available for ctenophore cells. (**B**) The stereotyped early development of *M. leidyi*.
The online version of this article includes the following figure supplement(s) for figure 1:

**Figure supplement 1.** Phylogenetic analysis for (**A**) *Ml*Par-6 and (**B**) *Ml*Par-1.
**Figure supplement 2.** Protein sequence alignment for *Ml*Par-6.
**Figure supplement 3.** Protein sequence alignment for *Ml*Par-1.

system components in ctenophores is important to understand the evolution of tissue organization in Metazoa.

The asymmetric localization of the Crumbs (Crb) complex, (e.g. Crb/Pals1/Patj), the Par/aPKC complex (e.g. Par-3/aPKC/Par-6), and the Scribble complex (e.g. Scribble/Lgl/Dlg) in the cortex of bilaterian and cnidarian cells maintains epithelial integrity by stabilizing cell-cell junctions (*Ohno et al., 2015*; *Salinas-Saavedra et al., 2015*; *Fahey and Degnan, 2010*; *Salinas-Saavedra et al., 2018*; *Belahbib et al., 2018*) via the Cadherin-Catenin complex (CCC) of mature Adherens Junctions (AJs) (*Magie and Martindale, 2008*; *Belahbib et al., 2018*; *Harris and Peifer, 2004*; *Nelson and Nusse, 2004*; *McGill et al., 2009*; *Oda and Takeichi, 2011*; *Schäfer et al., 2014*; *Weng and Wieschaus, 2016*). The maturation of AJs is essential for the maintenance of the Par/aPKC complex localization at the apical cortex that displaces members of the Scribble complex and Par-1 to basolateral localizations associated with Septate Junctions (SJs) (*Belahbib et al., 2018*; *Benton and St Johnston, 2003*; *Hurov et al., 2004*; *Zhang et al., 2007*; *Iden and Collard, 2008*;

*Yamanaka and Ohno, 2008*; *Oshima and Fehon, 2011*; *Ganot et al., 2015*; *Humbert et al., 2015*; *Kharfallah et al., 2017*).

   This mechanism is deployed in bilaterian cells to establish embryonic and epithelial cell polarity during early development and is critical for axial organization (*Salinas-Saavedra et al., 2015*; *Cha et al., 2011*; *Munro, 2006*; *Patalano et al., 2006*; *Goldstein and Macara, 2007*; *Weis-blat, 2007*; *Alford et al., 2009*; *Munro and Bowerman, 2009*; *Doerflinger et al., 2010*; *Chan and Nance, 2013*; *Lang and Munro, 2017*; *Tepass, 2012*; *Nance and Zallen, 2011*; *Weng and Wie-schaus, 2017*; *Zhu et al., 2017*; *Ragkousi et al., 2017*; *Salinas-Saavedra et al., 2018*; *Schneider and Bowerman, 2003*; *Macara, 2004*; *Vinot et al., 2004*; *Dollar et al., 2005*; *Ossipova et al., 2005*). Components of the Par system are unique to, and highly conserved, across Metazoa, including placozoans, poriferans, and ctenophores (*Fahey and Degnan, 2010*; *Belahbib et al., 2018*). But strikingly, ctenophore genomes do not have many of the crucial regula-tors present in other metazoan genomes (*Belahbib et al., 2018*; *Ganot et al., 2015*). For example, none of the components of the Crb complex, a Scribble homolog, or Human and *Drosophila* SJs, are present (*Belahbib et al., 2018*; *Ganot et al., 2015*), and the cytoplasmic domain of cadherin lacks the crucial biding sites to catenins that interact with the actin cytoskeleton (*Belahbib et al., 2018*). These data raise the question of whether or not ctenophore cells are polarized by mecha-nisms involving the apicobasal cell polarity mediated by Par proteins. Here, by using antibodies raised to specific ctenophore proteins and confirmed by live-cell imaging of injected fluorescently labeled mRNAs, we describe for the first time the subcellular localization of selected components of the Par system during the development of the ctenophore *Mnemiopsis leidyi*. Data obtained here challenge the conservation of the apicobasal cell polarity module and raise questions about the epi-thelial tissue organization as an evolutionary trait of all metazoans.

## Results

### *Ml*Par-6 gets localized to the apical cortex of cells during early *M. leidyi* development

We characterized the subcellular localization of the *Ml*Par-6 protein during early *M. leidyi* develop-ment by using our specific *Ml*Par-6 antibody (*Figure 2* and *Figure 2—figure supplements 1–6*). Although *Ml*Par-6 immunoreactivity can be detected in the periphery of the entire cell, in all of over 100 specimens examined, its expression appears to be polarized to the animal cortex (deter-mined by the position of the zygotic nucleus; *Figure 2A* and *Figure 2—figure supplements 8–10*) of the single cell zygote and to the apical (animal) cell cortex during every cleavage stage (*Fig-ure 2* and *Figure 2—figure supplement 3*). At the cortex, *Ml*Par-6 localizes to cell-contact-free regions facing the external media (*Figure 2C*). Gradually through the next three hours of develop-ment, *Ml*Par-6 becomes localized to the position of cell-cell contacts by 60 cell stage onwards (*Fig-ure 2—figure supplements 3E–G* and *4*). During gastrulation (3–7 hpf; *Figure 2D* and *Figure 2—figure supplements 3–4*), *Ml*Par-6 is not localized in cells undergoing cellular movements including the oral (four hpf; *Figure 2—figure supplement 3G*) and aboral ectoderm (5–6 hpf; *Figure 2D*) undergoing epibolic movements, syncytial endoderm, and mesenchymal 'mesoderm' (quotation marks its debatable homology). However, this protein remains polarized in 'static' ectodermal cells remaining at the animal pole (blastopore) and vegetal pole (4–7 hpf; *Figure 2—figure supplements 3F–J* and *4*). By the end of gastrulation (8–9 hpf; *Figure 2E*), *Ml*Par-6 becomes localized asymmetri-cally to the apical cortex of the ectodermal epidermal cells and the future ectodermal pharyngeal cells that start folding inside the blastopore (*Figure 2E* and *Figure 2—figure supplement 5A–C*). Interestingly, we do not observe a clear cortical localization in later cydippid stages, and the anti-body signal is weaker after 10 hpf in juveniles (*Figure 2F*). Contrary to expectations, at these later stages, *Ml*Par-6 is cytosolic and does not localize in the cortex of epidermal cells, and a few epithe-lial and mesenchymal cells showed nuclear localization (*Figure 2F*). Thereafter, *Ml*Par-6 remains cyto-solic in all scored stages up to 24 hpf (*Figure 2—figure supplement 6*). Cytosolic and nuclear localization of Par-6 has been reported in other organisms when the polarizing roles of this protein are inactive (*Mizuno et al., 2003*; *Johansson et al., 2000*; *Cline and Nelson, 2007*). Thus, our data suggest that *Ml*Par-6 does not play a role in cell polarity during juvenile cydippid stages. These pat-terns of apical localization seem not to be affected by the cell cycle (*Figure 2—figure supplements*

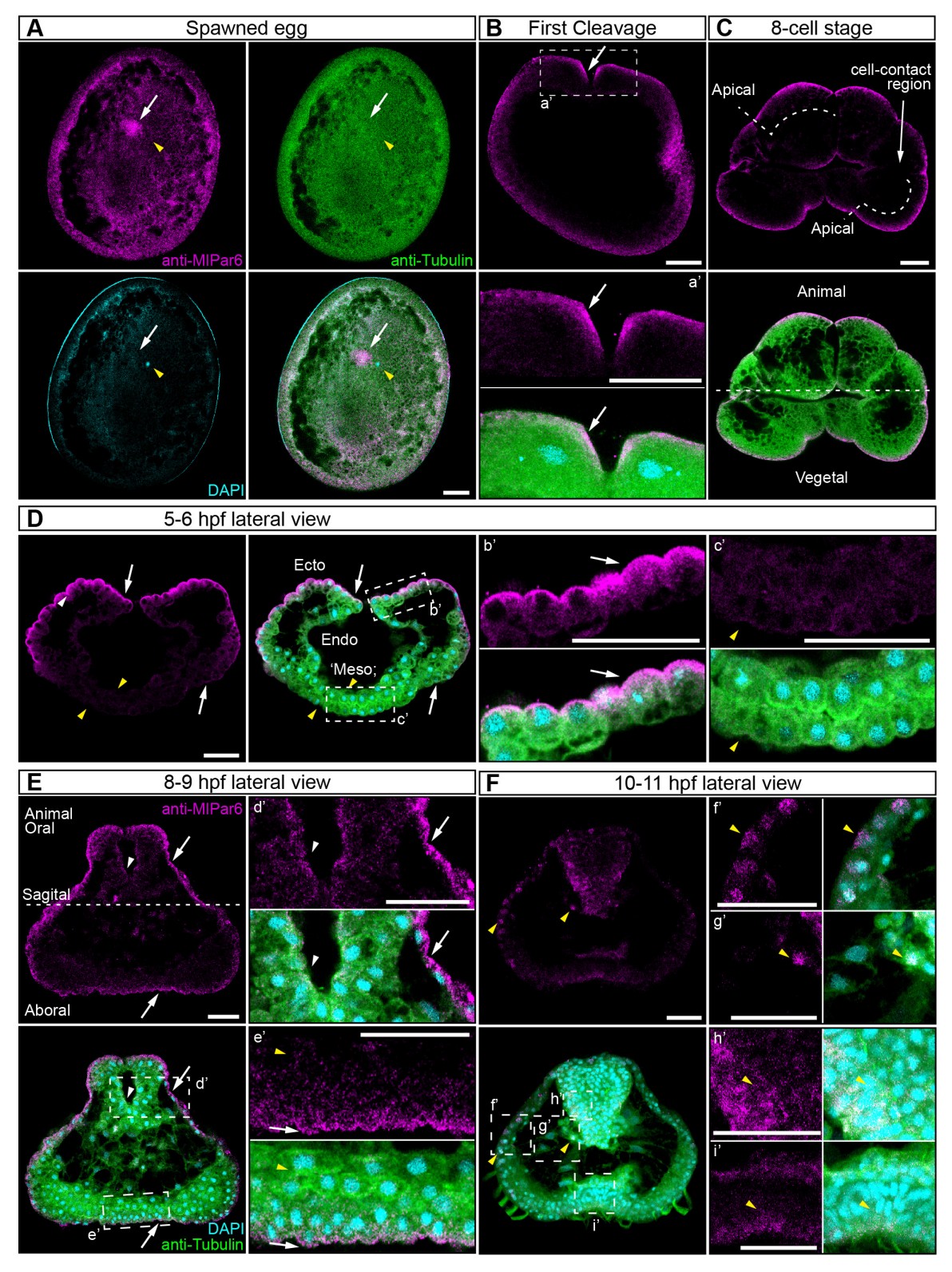

**Figure 2.** *Ml*Par-6 protein subcellular localization during the early development of *M. leidyi*. Immunostaining against *Ml*Par-6 protein shows that this protein localizes asymmetrically in the cell cortex of the eggs (**A**) and in the cell-contact-free regions of cleavage stages (**B–C**; white arrows). White circle in C indicates the lack of signal in the cell-contact region. Yellow arrowhead indicates the zygotic nucleus in A. a' is a magnification of the section depicted in (**B**) the first cleavage. (**D–F**): b' to i' correspond to magnifications of the regions depicted for each stage. (**D**) 5–6 hpf, *Ml*Par-6 protein

*Figure 2 continued on next page*

*Figure 2 continued*

localizes to the apical cortex of the ectodermal cells (Ecto) but is absent from endodermal (Endo) and 'mesodermal' ('Meso') cells. White arrowhead indicates *Ml*Par-6 protein in regions of cell-contact. Yellow arrowheads indicate the absence of cortical localization. (E) Until 9 hpf, *Ml*Par-6 protein localizes to the apical cortex of the ectoderm (white arrows) and pharynx (white arrowhead) but it is not cortically localized after 10 hpf (F; Yellow arrowheads indicate nuclear localization). Images are maximum projections from a z-stack confocal series. The 8 cell stage corresponds to a single optical section. Orientation axes are depicted in the Figure: Animal/oral pole is to the top. Morphology is shown by DAPI and Tubulin immunostainings. See *Figure 2—figure supplements 1–11* for expanded developmental stages. Scale bars: 20 μm.

The online version of this article includes the following source data and figure supplement(s) for figure 2:

**Figure supplement 1.** Diagram depicting the cortical localization of *Ml*Par-6 (magenta).
**Figure supplement 2.** Specificity of *M. leidyi* antibodies as tested by pre-adsorption experiments.
**Figure supplement 3.** *Ml*Par-6 localization during early developmental stages.
**Figure supplement 4.** *Ml*Par-6 localization during late gastrulation stages.
**Figure supplement 5.** *Ml*Par-6 localization during late developmental stages.
**Figure supplement 6.** Immunofluorescent staining against *Ml*Par-6 after 20 hpf.
**Figure supplement 7.** Schematic depiction of fluorescent intensity measurements correspondent to *Figure 2*.
**Figure supplement 8.** Fluorescent intensity measurements of immunofluorescent staining against *Ml*Par-6.
**Figure supplement 8—source data 1.** Numerical data that are represented as a graph in *Figure 2—figure supplement 8*.
**Figure supplement 9.** Fluorescent intensity distribution of immunofluorescent staining against *Ml*Par-6.
**Figure supplement 9—source data 1.** Numerical data that are represented as a graph in *Figure 2—figure supplement 9*.
**Figure supplement 10.** Graphical depiction of fluorescence intensity measurements between basal and apical cortex.
**Figure supplement 10—source data 1.** Numerical and statistical data that are represented as graphs in *Figure 2—figure supplement 10*.
**Figure supplement 11.** Fluorescent intensity measurements of immunofluorescent staining against *Ml*Par-6 during cell cycle.
**Figure supplement 11—source data 1.** Numerical data that are represented as a graph in *Figure 2—figure supplement 11*.
**Figure supplement 12.** Western blot analyses for the tested antibodies.

*8–11*). Further work is required to assess the relationship between cell cycle and the localization of these proteins.

Similar results were obtained when we overexpressed the mRNA encoding for *Ml*Par-6 fused to mVenus (*Ml*Par-6-mVenus) and recorded the *in vivo* localization of the protein in *M. leidyi* embryos (*Figure 2—figure supplement 5D–H*). Translated *Ml*Par-6-mVenus was observed approximately 4 hr post injection into the uncleaved egg so localization during early cleavage stags was not possible. However, during gastrulation, *Ml*Par-6-mVenus localizes to the apical cell cortex and displays enrichment at the level of cell-cell contacts (*Figure 2—figure supplement 5D–F*). As we observed by antibody staining, this cortical localization is no longer observable during the cell movements associated with gastrulation and *Ml*Par-6-mVenus remains cytosolic (*Figure 2—figure supplement 5D* bottom). After eight hpf, *Ml*Par-6-mVenus localizes to the apical cortex of ectodermal epidermal and pharyngeal cells but is not observable in any other internal tissue (*Figure 2—figure supplement 5G*). After 10 hpf, *Ml*Par-6-mVenus remains in the cytosol and no cortical localization was detectable (*Figure 2—figure supplement 5H*). Microinjection and mRNA expression in ctenophores is really challenging. For the first time, we have overexpressed fluorescent-tagged proteins for *in vivo* imaging. In spite of the low number of replicates (see Materials and methods), our results are consistent with the antibody observations presented above.

## *Ml*Par-1 remains cytoplasmic during early *M. leidyi* development

In bilaterians and cnidarians, the apical localization of *Ml*Par-6 induces the phosphorylation of *Ml*Par-1, displacing this protein to basolateral cortical regions (*Ohno et al., 2015*; *Salinas-Saavedra et al., 2015*; *Ragkousi et al., 2017*; *Salinas-Saavedra et al., 2018*). Using our specific *Ml*Par-1 antibody, we characterized the subcellular localization of the *Ml*Par-1 protein during the early *M. leidyi* development (*Figure 3* and all its supplements). Even though *Ml*Par-1 appears to be localized in the cortex at the cell-contact regions of early blastomeres and gastrula stages (*Figure 3D–E*), this antibody signal was not clear enough to be discriminated from the cytosolic distribution, possibly due to edge effects. Nevertheless, and strikingly, *Ml*Par-1 remains as punctate aggregations distributed uniformly in the cytosol, and in some cases, co-distributes with chromosomes during mitosis (*Figure 3* and *Figure 3—figure supplement 2*). We did not observe asymmetric localization of *Ml*Par-1 in the cell cortex of *M. leidyi* embryos at any of the stages described above for *Ml*Par-6.

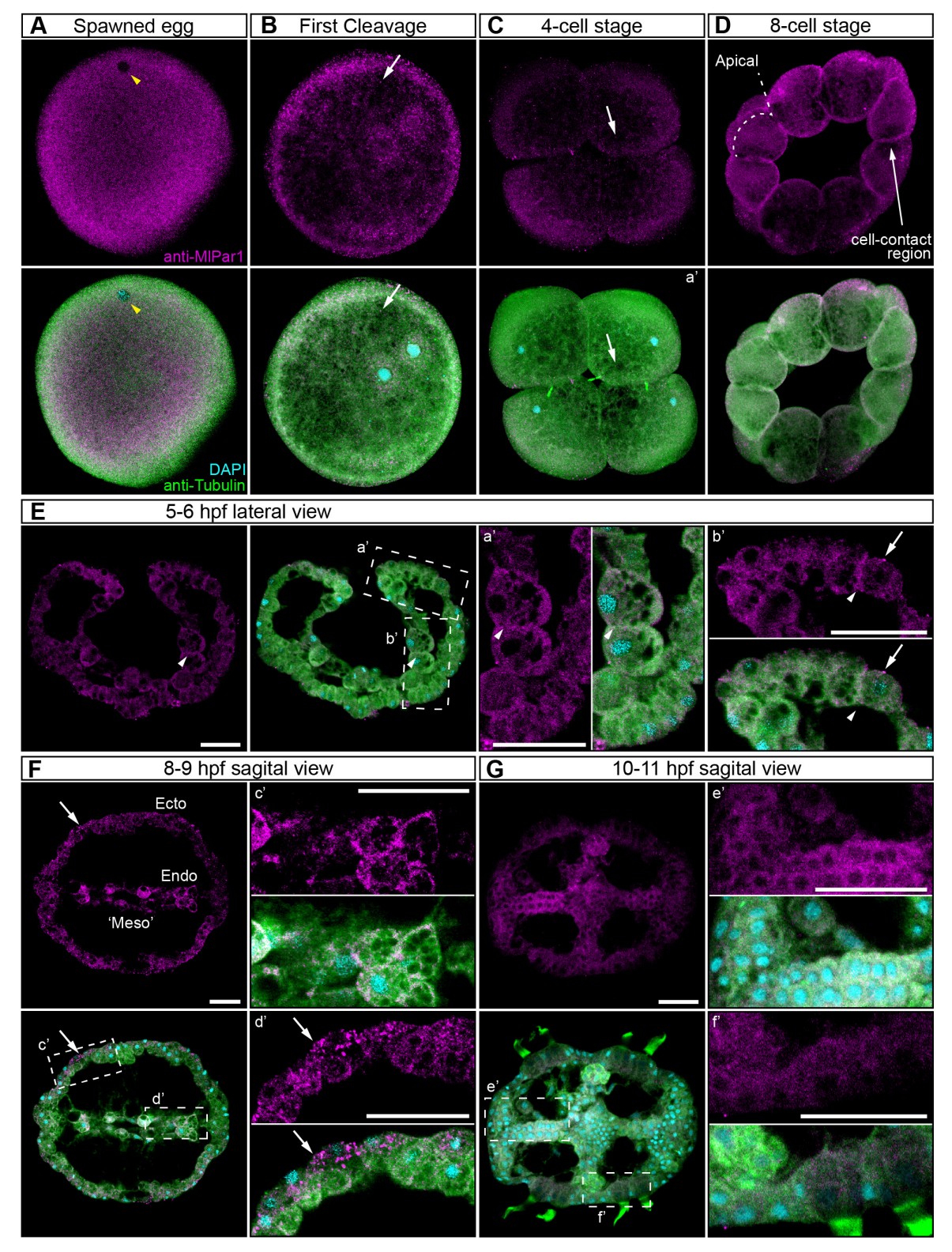

**Figure 3.** *Ml*Par-1 protein subcellular localization during the early development of *M. leidyi*. Immunostaining against *Ml*Par-1 protein shows that this protein remains cytoplasmic during early cleavage stages (**A–D**). *Ml*Par-1 protein appears as punctate aggregations distributed uniformly in the cytosol (white arrows). Yellow arrowhead indicates the zygote nucleus in (**A**). 8 cell-stage (**D**): A single optical section from a z-stack confocal series. *Ml*Par-1 appears to be localized in the cortex at the cell-contact regions but this antibody signal was similar to its cytosolic distribution. (**E–G**) Between 5 and 11

*Figure 3 continued on next page*

*Figure 3 continued*

hpf, *Ml*Par-1 protein remains as punctate aggregations distributed uniformly in the cytosol (white arrows). a' to f' correspond to the magnifications of the regions depicted for each stage. (E) *Ml*Par-1 appears to be localized in the cortex at the cell-contact regions (white arrowheads) but this antibody signal was similar to its cytosolic distribution. (F) *Ml*Par-1 protein remains cytoplasmic in ectodermal cells (Ecto; c'), endodermal (Endo; d'), and 'mesodermal' ('Meso') cells. Images are maximum projections from a z-stack confocal series. Sagittal view of an 8–9 hpf embryo corresponds to a single optical section from a z-stack confocal series. Orientation axes are depicted in the figure. Morphology is shown by DAPI and tubulin immunostainings. The animal pole is towards the top. Scale bars: 20 μm.

The online version of this article includes the following source data and figure supplement(s) for figure 3:

**Figure supplement 1.** Diagram depicting the cortical localization of *Ml*Par-1 (magenta).

**Figure supplement 2.** *Ml*Par-1 localization during developmental stages complementary to *Figure 3*.

**Figure supplement 3.** *Ml*Par-1 protein remains cytoplasmic during *M. leidyi* development between 8 hpf and 11 hpf.

**Figure supplement 4.** Immunofluorescent staining against *Ml*Par-1 after 20 hpf.

**Figure supplement 5.** Fluorescent intensity measurements correspondent to *Figure 3*.

**Figure supplement 5—source data 1.** Numerical data that are represented as graphs in *Figure 3—figure supplement 5*.

**Figure supplement 5—source data 2.** Numerical and statistical data that are represented as graphs in *Figure 3—figure supplement 5*.

**Figure supplement 6.** Schematic depiction of fluorescent intensity measurements correspondent to *Figure 3*.

These results were also supported *in vivo* when we overexpressed the mRNA encoding for *Ml*Par-1 fused to mCherry (*Ml*Par-1-mCherry) into *M. leidyi* embryos by microinjection (*Figure 3—figure supplement 3*). Similar to *Ml*Par-6-mVenus mRNA overexpression, the *Ml*Par-1-mCherry translated protein was observed after 4 hr post injection into the uncleaved egg. Our *in vivo* observations on living embryos confirm the localization pattern described above by using *Ml*Par-1 antibody at gastrula stages. *Ml*Par-1-mCherry localizes uniformly and form aggregates in the cytosol during gastrulation (4–5 hpf; *Figure 3—figure supplement 3D–E* and *Video 1*). This localization pattern remains throughout all recorded stages until cydippid juvenile stages where *Ml*Par-1-mCherry remains cytosolic in all cells but is highly concentrated in the tentacle apparatus and underneath the endodermal canals (24 hpf; *Figure 3—figure supplement 3F–G*, *Figure 3—figure supplement 4*, and *Video 2*).

## *Ml*Par-6 and *Ml*Par-1 Proteins can localize like host proteins localize in a heterologous system

To discount the possibility that the observations recorded *in vivo* for both *Ml*Par-6-mVenus and *Ml*Par-1-mCherry proteins are caused by a low-quality mRNA or lack of structural conservation, we overexpressed each ctenophore mRNA into embryos of the cnidarian *Nematostella vectensis* and followed their localization by *in vivo* imaging (*Figure 4*). In *N. vectensis* embryos, *Ml*Par-6-mVenus and *Ml*Par-1-mCherry symmetrically distribute during early cleavage stages (*Figure 4A and C*) and both proteins localize asymmetrically only after blastula formation (*Figure 4B and D*). In these experiments, both *Ml*Par-6-mVenus and *Ml*Par-1-mCherry translated proteins display the same pattern as the previously described endogenous *N.*

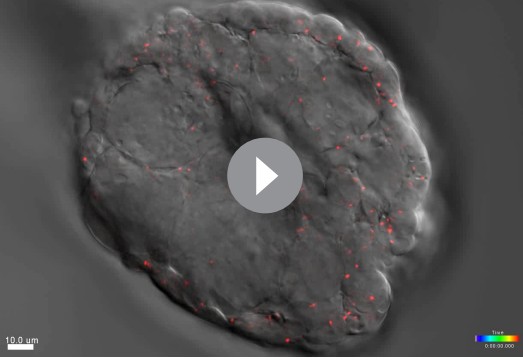

**Video 1.** Punctuate aggregates of *Ml*Par-1-mCherry are highly dynamic. 2.5 min *in vivo* recording of a gastrula embryo at 40x.

https://elifesciences.org/articles/54927#video1

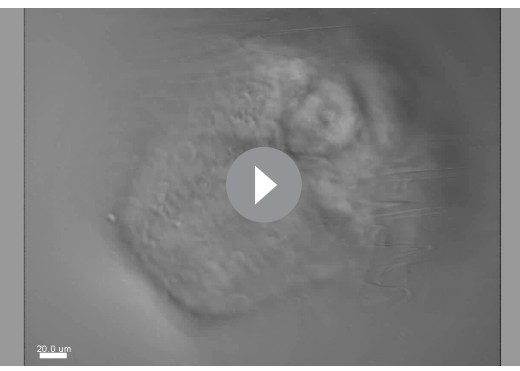

**Video 2.** Z-stack of *Ml*Par-1-mCherry expression at 24 hpf at 40X.

https://elifesciences.org/articles/54927#video2

*vectensis* Par-6 and Par-1 proteins (*Salinas-Saavedra et al., 2015*). These data suggest that the protein structure of ctenophore *Ml*Par-6 and *Ml*Par-1 contains the necessary information to localize as other bilaterians proteins do.

## Discussion

### Par protein asymmetry is established early but not maintained during *M. leidyi* embryogenesis

The asymmetric localization of the Par/aPKC complex has been used as an indicator of apical-basal cell polarity in a set of animals, including bilaterians (*Ohno et al., 2015*; *Salinas-Saavedra et al., 2015*; *Besson et al., 2015*; *Yang and Mlodzik, 2015*; *Goldstein and Macara, 2007*; *Munro and Bowerman, 2009*; *Doerflinger et al., 2010*; *Chan and Nance, 2013*; *Lang and Munro, 2017*; *Mizuno et al., 2003*; *Kemphues et al., 1988*; *Etienne-Manneville and Hall, 2003*; *Vinot et al., 2005*; *Lee et al., 2007*; *Martindale and Hejnol, 2009*; *Martindale and Lee, 2013*; *Chalmers et al., 2005*; *Hayase et al., 2013*) and a cnidarian (*Salinas-Saavedra et al., 2015*; *Ragkousi et al., 2017*). While in the studied bilaterians this asymmetry is established and maintained since the earliest stages of development (*Munro and Bowerman, 2009*; *Lang and Munro, 2017*; *Zhu et al., 2017*; *Nance, 2014*; *Hoege and Hyman, 2013*; *Von Stetina and Mango, 2015*), in the cnidarian *N. vectensis* there is no early asymmetrical localization of any of the Par components (*Salinas-Saavedra et al., 2015*; *Ragkousi et al., 2017*) and embryonic polarity is controlled by the Wnt signaling system (*Kumburegama et al., 2011*; *Wikramanayake et al., 2003*; *Lee et al., 2007*; *Martindale and Hejnol, 2009*; *Martindale and Lee, 2013*). In spite of these differences, once epithelial tissues form and epithelial cell-polarity is established in both bilaterian and cnidarian species, the asymmetric localization of Par proteins become highly polarized and is maintained through development. In those cases, Par-mediated apicobasal cell polarity is responsible for the maturation and maintenance of cell-cell adhesion in epithelial tissue (*Ohno et al., 2015*; *Salinas-Saavedra et al., 2018*). We have suggested that the polarizing activity of the Par system was already present in epithelial cells of the MRCA between Bilateria and Cnidaria (*Salinas-Saavedra and Martindale, 2018*; *Salinas-Saavedra and Martindale, 2018*) and could be extended to all Metazoa, where these proteins are present (including ctenophores, sponges, and placozoans *Fahey and Degnan, 2010*; *Belahbib et al., 2018*).

However, our current data suggest a different scenario for ctenophores where the Par protein polarization observed during earlier stages (characterized by the apical and cortical localization of *Ml*Par-6; *Figure 2*) is not maintained when ctenophore juvenile epithelial tissues form after nine hpf. Epithelial cells of later cydippid stages do not display an asymmetric localization of *Ml*Par-6 (*Figure 2—figure supplement 6*). Furthermore, the subcellular localization of *Ml*Par-1 does not display a clear localization during any of the observed developmental stages (*Figure 3* and all its supplements). Instead, punctate aggregates distribute symmetrically in the cytosol. *Ml*Par-1 and mCherry aggregates may be consequence of the highly protein availability in the cytosol that is not captured to the cell cortex.

The components of the ctenophore *Ml*Par/aPKC complex (*Ml*Par-3/*Ml*aPKC/*Ml*Par-6 and *Ml*Cdc42) are highly conserved and contain all the domains present in other metazoans (*Figure 1—figure supplements 1–2*; *Fahey and Degnan, 2010*; *Belahbib et al., 2018*). Similarly, the primary structure of *Ml*Par-1 protein (a Serine/threonine-protein kinase) is highly conserved and contains all the domains (with the same amino acid length) required for its proper functioning in other metazoans (*Figure 1—figure supplement 3*; *Fahey and Degnan, 2010*; *Belahbib et al., 2018*), and localizes to the lateral cortex when expressed in cnidarian embryos (*Figure 4*). Regardless, these proteins do not asymmetrically localize to the cortex of *M. leidyi* juvenile epithelium. Interestingly, the punctuate aggregates of *Ml*Par-1-mCherry are highly dynamic and move throughout the entire cytosol (*Figure 3—figure supplement 3*), suggesting a potential association with cytoskeletal components (see *Video 1*) as *Ml*Par-1 conserve these motifs.

Recent studies have shown that ctenophores do not have homologs for any of the Crb complex components (*Belahbib et al., 2018*), required for the proper stabilization of the CCC and Par/aPKC complex in other studied taxa (*Ohno et al., 2015*; *Harris and Peifer, 2004*; *Tepass, 2012*; *Chalmers et al., 2005*; *Hayase et al., 2013*; *Whitney et al., 2016*). The lack of *Ml*Par-6 (*Figure 2*)

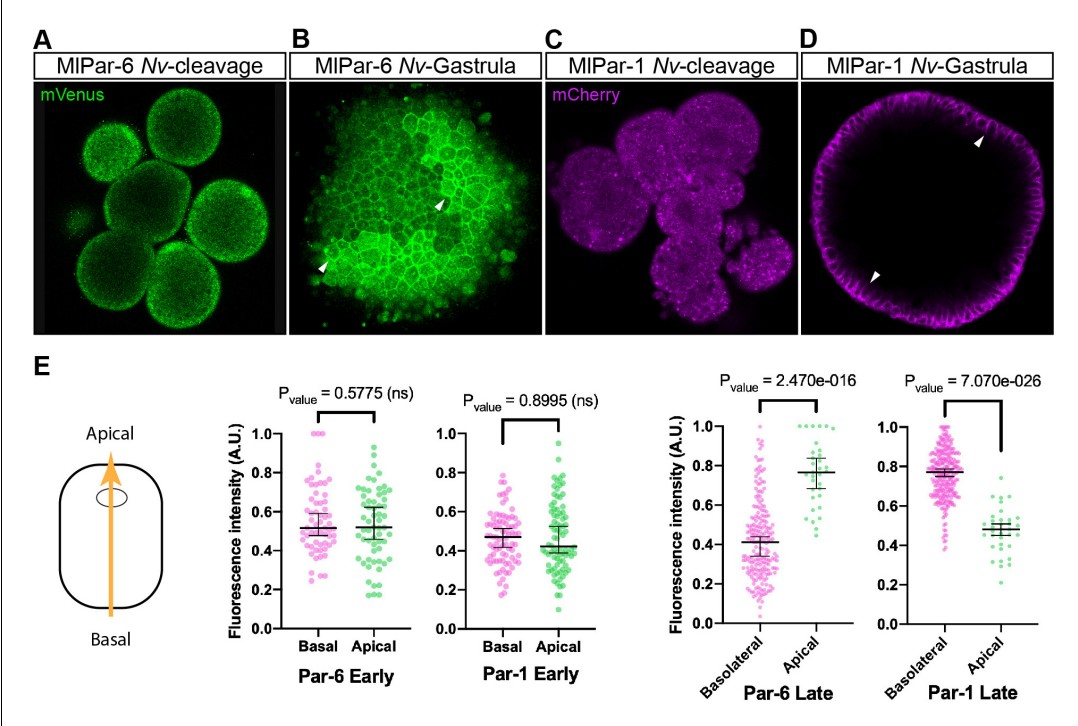

**Figure 4.** Expression of ctenophore *Ml*Par6-mVenus and *Ml*Par1-mCherry in embryos of the cnidarian *N. vectensis*. The translated exogenous proteins display the same pattern than the previously described for endogenous *N. vectensis* proteins (A–D). White arrowheads indicate *Ml*Par6-mVenus and *Ml*Par1-mCherry cortical localization (B and D). All images are a single slice from a z-stack confocal series. (E) Graphical depiction of fluorescence intensity measurements between basal and apical cortex. The diagram at the left shows the direction of the measurements represented in this figure and in *Figure 4—figure supplement 2*. Median, 95% CI, and P values are depicted in the figure.

The online version of this article includes the following source data and figure supplement(s) for figure 4:

**Figure supplement 1.** Evolution of cell polarity in Metazoa.

**Figure supplement 2.** Fluorescent intensity measurements correspondent to *Figure 4*.

**Figure supplement 2—source data 1.** Numerical data that are represented as graphs in *Figure 4—figure supplement 2*.

**Figure supplement 2—source data 2.** Numerical and statistical data that are represented as graphs in *Figure 4—figure supplement 2*.

polarization during later stages is totally congruent with these observations, indicating that Par proteins in ctenophores do not have the necessary interactions to stabilize apico-basal cell polarity in their cells as in other animals. In addition, ctenophore species do not have the molecular components to form SJs and lack a Scribble homolog (*Belahbib et al., 2018*; *Ganot et al., 2015*). This could explain the cytosolic localization of *Ml*Par-1 during the observed stages (*Benton and St Johnston, 2003*; *Iden and Collard, 2008*; *Humbert et al., 2015*; *Bilder et al., 2000*; *Vaccari et al., 2005*), (*Bonello et al., 2019*).

## Evolution of cell polarity and epithelial structure in metazoa

Given the genomic conservation of cell-polarity components in the Bilateria and Cnidaria, we propose to classify their epithelium as 'Par-dependent' to include its mechanistic regulatory properties. That is, the structural properties of a 'Par-dependent'epithelium are the result of conserved interactions between subcellular pathways that polarize epithelial cells. Thus, when we seek to understand the origins of the epithelial nature of one particular tissue, we are trying to understand the synapomorphies (shared derived characters) of the mechanisms underlying the origin of that particular tissue. Under this definition, a 'Par-dependent epithelium' may have a single origin in Metazoa, but, different mechanisms might have co-opted to generate similar epithelial morphologies (*Figure 4—figure supplement 1*). Ctenophore epithelia, along with other recent works in *N. vectensis* endomesoderm (*Salinas-Saavedra et al., 2015*; *Salinas-Saavedra et al., 2018*) and *Drosophila* midgut (*Chen et al., 2018*), suggest this possibility. In all these cases, epithelial cells are highly polarized

along the apical-basal axis, but this polarization does not depend on Par proteins. Therefore, these cells are not able to organize a 'Par-dependent epithelium' (mechanistic definition) but still polarized epithelial morphologies.

Genomic studies also suggest that ctenophore species lack the molecular interactions necessary to form the apical cell polarity and junctions observed in Cnidaria + Bilateria. Intriguingly, ctenophore genomes do not have the Wnt signaling pathway components (*Ryan et al., 2013*; *Moroz et al., 2014*; *Pang et al., 2010*) that control the activity of Par proteins in bilaterian and cnidarian embryos (components that are also present in poriferan and placozoan genomes *Belahbib et al., 2018*). For example, in bilaterians the Wnt/PCP signaling pathway antagonizes the action of the Par/aPKC complex (*Cha et al., 2011*; *Besson et al., 2015*; *Aigouy and Le Bivic, 2016*; *Humbert et al., 2015*; *Humbert et al., 2006*; *Seifert and Mlodzik, 2007*), so this may explain the lack of polarization in ctenophore tissue. Furthermore, ctenophore species do not have the full set of cell-cell adhesion proteins (*Belahbib et al., 2018*; *Ryan et al., 2013*; *Ganot et al., 2015*) as we know them in other metazoans, including Placozoans and Poriferans (*Magie and Martindale, 2008*; *Belahbib et al., 2018*). The cadherin of ctenophores does not have the cytoplasmic domains required to bind any of the catenins of the CCC (e.g. p120, alpha- and ß-catenin) (*Belahbib et al., 2018*). This implies that neither the actin nor microtubule cytoskeleton can be linked to ctenophore cadherin through the CCC, as seen essential in other metazoans to stabilize pre-existent Par proteins polarity. This suggests that there are additional mechanisms that integrate the cytoskeleton of ctenophore cells with their cell-cell adhesion system.

In conclusion, regardless the phylogenetic position of the Ctenophora, the conservation of an organized 'Par-dependent epithelium' cannot be extended to all Eumetazoa. Ctenophore cells do not have other essential components to organize the polarizing function of the Par system as in other studied metazoans. Despite the high structural conservation of Par proteins across Metazoa, we have shown that ctenophore cells do not deploy and/or stabilize the asymmetrical localization of Par-6 and Par-1 proteins. Thus, ctenophore tissues organize their epithelium in a different way than the classical definition seen in bilaterians. In agreement with genomic studies, our results question what molecular properties defined the ancestral roots of a metazoan epithelium, and whether similar epithelial morphologies (e.g., epidermis and mesoderm) could be developed by independent or modifications of existing cellular and molecular interactions (including cell adhesion systems). Unless the lack of Par protein localization in *M. leidyi* is a secondary loss, the absence of these pathways in ctenophores implies that a new set of interactions emerged at least in the Cnidaria+Bilateria ancestor (*Figure 4—figure supplement 1*), and that, could have regulated the way by which the Par system polarizes embryonic and epithelial cells. While bioinformatic studies are critical to understand the molecular composition, we need further research to understand how these molecules actually interact with one another to organize cellular behavior (e.g., integrin-collagen, basal-apical interactions) in a broader phylogenetical sample, including Porifera and Placozoa.

# Materials and methods

**Key resources table**

| Reagent type (species) or resource | Designation | Source or reference | Identifiers | Additional information |
|---|---|---|---|---|
| Antibody | Mouse Anti-alpha-Tubulin Monoclonal Antibody, Unconjugated, Clone DM1A | Sigma-Aldrich | T9026; RRID:AB_477593 | (1:500) |
| Antibody | anti-MlPar-6 custom peptide antibody produced in rabbit | Bethyl labs; This study | | Stored at MQ Martindale's lab; (1:100) |
| Antibody | anti-MlPar-1 custom peptide antibody produced in rabbit | Bethyl labs; This study | | Stored at MQ Martindale's lab; (1:100) |

*Continued on next page*

*Continued*

| Reagent type (species) or resource | Designation | Source or reference | Identifiers | Additional information |
|---|---|---|---|---|
| Antibody | Goat anti-Mouse IgG Secondary Antibody, Alexa Fluor 568 | Thermo Fisher Scientific | A-11004; RRID:AB_2534072 | (1:250) |
| Antibody | Goat anti-Rabbit IgG Secondary Antibody, Alexa Fluor 647 | Thermo Fisher Scientific | A-21245; RRID:AB_2535813 | (1:250) |
| Other | DAPI (4',6-Diamidino-2-Phenylindole, Dihydrochloride) | Thermo Fisher Scientific | D1306; RRID:AB_2629482 | (0.1 µg/µl) |
| Chemical compound, drug | Dextran, Alexa Fluor 488; 10,000 MW, Anionic, Fixable | Thermo Fisher Scientific | D22910 | |
| Chemical compound, drug | Dextran, Alexa Fluor 555; 10,000 MW, Anionic, Fixable | Thermo Fisher Scientific | D34679 | |
| Chemical compound, drug | Dextran, Alexa Fluor 647; 10,000 MW, Anionic, Fixable | Thermo Fisher Scientific | D22914 | |
| Chemical compound, drug | Dextran, Cascade Blue, 10,000 MW, Anionic, Lysine Fixable | Thermo Fisher Scientific | D1976 | |
| Sequence-based reagent | *Mlpar-6*: F-GTACTGTGC TGTGTGTTTGGA; R- GTACTGTGCT GTGTGTTTGGA | *Mnemiopsis Genome* Project - NIH-NHGRI | MLRB351777 | |
| Sequence-based reagent | *Mlpar-1*: F- ATGTCAAA TTCTCAACACCAC; R- CAGTCTTAATTCA TTAGCTATGTTA | *Mnemiopsis Genome* Project - NIH-NHGRI | MLRB182569 | |
| Recombinant DNA reagent | pSPE3-mVenus | *Roure et al., 2007* | | Gateway vector |
| Recombinant DNA reagent | pSPE3-mCherry | *Roure et al., 2007* | | Gateway vector |
| Software, algorithm | Fiji (ImageJ) | NIH | http://fiji.sc | |
| Software, algorithm | Imaris 7.6.4 | Bitplane Inc | | |

## Culture and spawning of *M. leidyi*

Spawning, gamete preparation, fertilization and embryo culturing of *M. leidyi* at the Whitney Laboratory for Marine Bioscience of the University of Florida (USA)embryos was performed as previously described (*Salinas-Saavedra and Martindale, 2018*).

## Western blot

Western blots were carried out as described (*Salinas-Saavedra et al., 2015*; *Salinas-Saavedra et al., 2018*) using adult epithelial tissue lysates dissected by hand in order to discard larger amount of mesoglea. Antibody concentrations for Western blot were 1:1000 for all antibodies tested.

## Immunohistochemistry

All immunohistochemistry experiments were carried out using the previous protocol for *M. leidyi* (*Salinas-Saavedra and Martindale, 2018*). The primary antibodies and concentrations used were:

mouse anti-alpha tubulin (1:500; Sigma-Aldrich, Inc Cat.# T9026. RRID:AB_477593). Secondary antibodies are listed in the Key Resources table. Rabbit anti-*Ml*Par-6, and rabbit anti-*Ml*Par-1 antibodies were custom made high affinity-purified peptide antibodies that commercially generated by Bethyl labs, Inc (Montgomery, TX, USA). Affinity-purified *M. leidyi* anti-Par-6 (anti-*Ml*Par-6) and anti-Par-1 (anti-*Ml*Par-1) peptide antibodies were raised against a selected amino acid region of the *Ml*Par-6 protein (MTYPDDSNGGSGR) and *Ml*Par-1 protein (KDIAVNIANELRL), respectively. Blast searches against the *M. leidyi* genome sequences showed that the amino acid sequences were not present in any predicted *M. leidyi* proteins other than the expected protein. Both antibodies are specific to *M. leidyi* proteins (*Figure 2—figure supplement 2*) and were diluted 1:100.

## mRNA microinjections

The coding region for each gene of interest was PCR-amplified using cDNA from *M. leidyi* embryos and cloned into pSPE3-mVenus or pSPE3-mCherry using the Gateway system (*Roure et al., 2007*). To confirm the presence of the transcripts during *M. leidyi* development, we cloned each gene at 2 hpf and 48 hpf. *N. vectensis* eggs were injected directly after fertilization as previously described (*Salinas-Saavedra et al., 2015*; *DuBuc et al., 2014*; *Layden et al., 2013*) with the mRNA encoding one or more proteins fused in frame with reporter fluorescent protein (N-terminal tag) using an optimized final concentration of 300 ng/µl for each gene. Fluorescent dextran was also co-injected to visualize the embryos. Live embryos were kept at room temperature and visualized after the mRNA of the FP was translated into protein (4–5 hr). Live embryos were mounted in 1x sea water for visualization. Images were documented at different stages. We injected and recorded at least 20 embryos for each injected protein and confocal imaged each specimen at different stages for detailed analysis of phenotypes *in vivo*. We repeated each experiment at least five times obtaining similar results for each case. The fluorescent dextran and primers for the cloned genes are listed in Key resources table.

## Imaging of *M. leidyi* embryos

Images of live and fixed embryos were taken using a confocal Zeiss LSM 710 microscope using a Zeiss C-Apochromat 40x water immersion objective (N.A. 1.20). Pinhole settings varied between 1.2–1.4 A.U. according to the experiment. The same settings were used for each individual experiment to compare control and experimental conditions. Z-stack images were processed using Imaris 7.6.4 (Bitplane Inc) software for three-dimensional reconstructions and FIJI for single slice and videos. Final figures were assembled using Adobe Illustrator and Adobe Photoshop.

Par proteins display a general cytosolic localization when their polarizing activity is inactive. This signal was diminished by modifying contrast and brightness of the images in order to enlighten their cortical localization (active state in cell-polarity and stronger antibody signal) as it has shown in other organisms. All RAW images are available upon request.

## Fluorescent intensity measurements and statistical analyses

Images of fixed embryos were measured using FIJI plot profile tool using the RAW source data. Fluorescent intensity was measured along the animal-vegetal axis for 1 and 2 cell stages and along the apico-basal axis for the other later stages. The data obtained were then normalized by the maximum value of each X and Y axes. X axis corresponds to the distance from basal (0) to apical (1) cortex. Y axis corresponds to fluorescence intensity. The normalized data were plotted and the numerical values can be found in figure supplement-data source files. For later stages than 8 cells, we took measurements of two cells located in perpendicular axes of the embryo where the apico-basal axis was clearly detectable. These measurements correspond to cells going through interphase and metaphase. Statistical analyses were executed using GraphPad prism software. To do this, we compared the 10% most basal positions with the 10% most apical positions for each stage. We plotted this data and differences were assessed by comparing medians using Mann-Whitney U test.

Similarly, fluorescent intensity during cell cycle (*Figure 2—figure supplement 11*) was measured along the apical cortex. The data obtained were then normalized by the maximum value of each X and Y axes. X axis corresponds to the arbitrary distance (0 to 1) along the apical cortex where the middle point corresponds to the cell-cell contact region or cleavage furrow. Y axis corresponds to

fluorescence intensity. The normalized data were plotted and the numerical values can be found in *Figure 2—figure supplement 11—source data 1*.

## Acknowledgements

We thank J Torres-Paz, CE Schnitzler, U Frank, and J Ryan for technical assistance and helpful comments.

## Additional information

### Funding

| Funder | Grant reference number | Author |
| --- | --- | --- |
| National Science Foundation | NSF IOS-1755364 | Mark Q Martindale |
| National Aeronautics and Space Administration | NASA 16-EXO16_2-0041 | Mark Q Martindale |

The funders had no role in study design, data collection and interpretation, or the decision to submit the work for publication.

### Author contributions

Miguel Salinas-Saavedra, Conceptualization, Resources, Validation, Investigation, Visualization, Methodology, Writing - original draft, Project administration, Writing - review and editing; Mark Q Martindale, Conceptualization, Resources, Supervision, Funding acquisition, Writing - original draft, Project administration, Writing - review and editing

### Author ORCIDs

Miguel Salinas-Saavedra (iD) https://orcid.org/0000-0002-1598-9881

### Decision letter and Author response

Decision letter https://doi.org/10.7554/eLife.54927.sa1
Author response https://doi.org/10.7554/eLife.54927.sa2

## Additional files

### Supplementary files

• Transparent reporting form

### Data availability

Genomic and Sequencing data can be found in the Mnemiopsis Genome Project (NIH-NHGRI) web-page (http://kona.nhgri.nih.gov/mnemiopsis/). We used the genome and prediction models to search the sequences of Par6 (https://kona.nhgri.nih.gov/mnemiopsis/jbrowse/data.cgi?type=unfiltered2.2&gene=MLRB351777) and Par1 (https://kona.nhgri.nih.gov/mnemiopsis/jbrowse/data.cgi?type=unfiltered2.2&gene=MLRB182569) using the blast tool. All data generated or analyzed during this study are included in the manuscript and supporting files.

The following dataset was generated:

| Author(s) | Year | Dataset title | Dataset URL | Database and Identifier |
| --- | --- | --- | --- | --- |
| Salinas-Saavedra M, Martindale MQ | 2020 | Par protein localization during the early development of Mnemiopsis leidyi suggests different modes of epithelial organization in the Metazoa | https://www.ebi.ac.uk/biostudies/studies/S-BSST502 | BioStudies, S-BSST502 |

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
