## [Decision Letter]

**Acceptance summary:**

This work provides insight into the evolution of body plans by molecularly characterizing epithelial organization in the early branching ctenophore, *Mnemiopsis leidyi*, including the first in vivo cell imaging of proteins in a ctenophore embryo. These data provide new insight into how Par proteins might pattern epithelial tissue in animals.

**Decision letter after peer review:**

Thank you for submitting your article "Par protein localization during the early development of *M. leidyi* suggests different modes of epithelial organization" for consideration by *eLife*. Your article has been reviewed by two peer reviewers, including Alejandro Sánchez Alvarado as the Reviewing Editor and Reviewer #1, and the evaluation has been overseen by Patricia Wittkopp as the Senior Editor.

The reviewers have discussed the reviews with one another and the Reviewing Editor has drafted this decision to help you prepare a revised submission.

Summary:

This paper by Salinas-Saavedra and Martindale reports on a molecular characterization of epithelial organization in ctenophores using immunohistochemistry and expression of fusion mRNAs. The paper focuses on the endogenous, polarity-associated Par proteins Par-1 and Par-6 and comparing these to fusion mRNAs at the same developmental stages. The authors heterologously express ctenophore Par fusion mRNAs in the cnidarian, *Nematostella vectensis*. They show that while in *Nematostella*, ctenophore Par proteins localize in a similar fashion to endogenous cnidarian Par proteins, but that in the ctenophore, based on their localization patterns during embryonic development, they conclude that Par-6 localizes asymmetrically to the apical surface of blastomeres early in development but then later fails to localize in epithelia. They also show that ctenophore Par-1 is primarily localized to cytosolic puncta during most stages of development.

Essential revisions:

While the results are certainly intriguing, and we commend the authors for generating key reagents that will be critical in addressing the evolutionary origin of metazoan epithelium there are several major concerns that would need to be addressed before the work is suitable for publication. We list these below:

1) A more rigorous demonstration that the cell cycle is not affecting protein distribution is necessary. For example, evidence that mitotic cells may have different localization of Par-6 than during interphase is shown in the supplemental figure of MlPar-6 Ab staining showing 4 hpf Animal view. Here, Par-6 localization appears in two cells that are likely in anaphase and there is Par-6 localization polarized to opposite sides of the daughter cells.

2) Given the nature of the claims made in the paper, it is necessary to quantify the imaging data reported. Making a generalization of localization based on single representative images without quantification is not acceptable. While the authors claim in the body of the text (subsection “*Ml*Par-6 gets localized to the apical cortex of cells during early *M. leidyi* development”) to have looked at hundreds of embryos, it would make more sense to quantify the localization using FIJI/ImageJ at each developmental stage and then display the representative image that best shows the quantified data as well as include how many embryos were examined at each developmental stage. Without being able to examine all of the collected image data, it seems problematic to make a generalization of localization based on single representative images without quantification. For example, for the early blastomere localization of Par-6, it would be important to show the fluorescence intensity of the apical vs. basal side of the population of blastomeres to make a claim about the establishment of polarity, especially during the earliest cell divisions. By performing a quick line scan in FIJI of the data across the cells, it is not particularly convincing that the signal is polarized to the apical cortex. At first cleavage, there is just as much signal in the lateral right portion of the embryo as there is at the cytokinetic furrow.

3) It would also be helpful to include a maximum projection of the collected Z-stack of each representative image to highlight the localization of the proteins.

4) It is unclear how the Ragkousi et al., 2017 is considered in the schematic. Ragkousi et al., show that *Nematostella* embryos lose polarity and the localization of Par-6 only during mitotic rounding, yet re-establish polarity during each interphase when they reinforce cell-cell contacts and are compacted. As the authors have primarily examined static time points, it is important to take into account cell cycle state, particularly the difference between interphase and mitotic cells. For example, in many of the images shown in the figures the cells are undergoing mitosis. It might be important to compare localization between interphase cells. Data from *Nematostella* (Ragkousi et al., 2017) demonstrates that localization changes of Par proteins between interphase and mitosis during early cleavages of blastomere (more on the interpretation of this data below). It would be extremely interesting to know if something similar is occurring during early *Mnemiopsis* cell divisions. One striking example where mitotic cells may have different localization of Par-6 than during interphase is the image from the supplement showing Par-6 localization where to the left and right of center there are two cells that appear to be in anaphase and there is Par-6 localization polarized to opposite sides of the daughter cells (supplemental figure of MlPar-6 Ab staining showing 4 hpf Animal view).

5) In reference to Figure 2—figure supplement 4B which shows the localization of MIPar-6-mVenus – it is hard to see how similar this localization pattern is to the endogenous Par6. The authors should fix the MIPar-6-mVenus expressing embryos and stain for Par6. For example, the antibody staining for Par6 at 5hpf shows an enrichment in the cells on the animal pole, but this localization is not shown with the MIPar-6-mVenus expressing animals.

6) In reference to the summary model – it is unclear how interpretation of the Ragkousi et al., paper is considered in the schematic. In their paper, they showed that *Nematostella* embryos lose polarity and the localization of Par-6 only during mitotic rounding but re-establish polarity during each interphase when they reinforce cell-cell contacts and are compacted. It would be extremely interesting to know if something similar is occurring during *Mnemiopsis* cell divisions early. Also, it would be worth more carefully visualizing the heterologous experiments or including images of compacted *Nematostella* embryos expressing ctenophore Par-6 to verify that it localizes in the same manner as the endogenous NvPar-6.

[Editors' note: further revisions were suggested prior to acceptance, as described below.]

Thank you for submitting your article "Par protein localization during the early development of *Mnemiopsis leidyi* suggests different modes of epithelial organization in the Metazoa" for consideration by *eLife*. Your article has been reviewed by Patricia Wittkopp as the Senior Editor, a Reviewing Editor, and two reviewers. The reviewers have opted to remain anonymous.

The reviewers have discussed the reviews with one another and the Reviewing Editor has drafted this decision to help you prepare a revised submission.

We thank the authors for preparing this revised version of their work showing that Par6 polarizes in early but not late blastomeres. The manuscript was re-evaluated by one of the original reviewers (reviewer #2) and a new reviewer (reviewer #3), who prepared their independent thoughts and then evaluated the prior reviews and the authors' response to reviewer. As you will see below, reviewer #2 still has some remaining concerns, particularly about statistical analysis and the need for a schematic, whereas reviewer #3 has more extensive concerns about over-interpretation in the manuscript. In the post review discussion, reviewer #2 agreed that reviewer #3's concerns are valid and that they would also like to see them addressed. I am therefore inviting a further revised version (which is rarely done at *eLife*) so that you can address these issues. The reviewers agree that no new data is required. Specifically, they think that their "comments could be addressed by text revisions, especially toning down the overly strong conclusions. The authors also need to show some sequence alignments, and ideally also domain prediction, to support their claims about the conservation of Ctenophore PAR proteins; but this would only require computational work rather than new experiments."

The full set of comments from both reviewers appear below.

Reviewer #2:

We commend the authors for responding to our previous comments. We also thank the authors for clarifying that ctenophore interphase cells maintain mid-bodies and apologize for a lack of familiarity on these fascinating embryos. In regards to the addition of the requested quantification of their data, we have several concerns (specifics detailed below). Anytime quantification is done, it is appropriate to also include statistical tests to support claims (of either no change in localization across an area measured or a significant change across the area measured). Additionally, it would be very helpful to the readers to include a schematic of where the line scans were done – either utilizing the staged drawings of the ctenophore embryos or placing a line on an example micrograph.

Specifically:

Figure 2—figure supplement 7 – Line scans appear very noisy for 1 and 2 cell stage measurements (i.e., there is a large degree of variability between the 4 quantified embryos at the 1-cell stage and 1/3 embryos at the 2-cell stage is quite different than the other 2) – at the 4 cell stage there seems to be a clear enrichment on apical vs basal, but this would require statistical tests to make the appropriate claim. – or relative enrichment on apical vs basal would make this quantitatively clear and allow the authors to make (or not) their claims.

Figure 2—figure Supplement 8

Here, the argument for telophase enrichment only has an n=2 embryos for both 2 and 4 cell stages – in which one embryo of the 2 does seem to have a dramatic changes in enrichment along the apical surface. We recognize the difficulties associated with imaging these embryos but the data needs to be consistent across samples to make claims, even where statistics may not be able to be used due to low sample size (but some sort of claim needs to be made stating that a statistical analysis could not be performed due to this low (n) issue).

Additionally, we had requested that the authors examine apical vs. basal enrichment during the cell cycle, but it appears that they have quantified the distribution of signal across the apical cortex only, but for example at first cleavage there is a dip in signal at the mid-point of their line scan, but in the text they make the conclusion that there is more Par6 at the cleavage furrow – or is the dip in signal overlapping with the absence of localization right at the middle of the cleavage furrow (Figure 2B/A'). Again, a schematic or annotation of how the line scans were made would be extremely useful for interpretation of data.

Figure 3 and Figure 4 – Really, to make any claims about localization, the quantification should be included in the main figures as well as an annotation of where the line scan was made in a representative image along with reporting the (n) analyzed and the appropriate statistical tests included and referenced in the main text manuscript.

Reviewer #3:

It is widely accepted that polarized epithelial tissues are a conserved feature of all metazoans, although the vast majority of studies have focused on bilaterian model organisms. Here, Salinas-Saavedra and Martindale examine cell polarity in *M. leidyi*, a representative of the basally branching ctenophore lineage. Using immunostaining and mRNA injections, the authors find that two well-studied polarity proteins, Par-6 and Par-1, exhibit a polarized localization in a smaller number of cells and during a narrower range of times during *M. leidyi* embryogenesis than might have been predicted based on studies in other systems.

The examination of epithelial polarity in an early-branching metazoan lineage is interesting and important, and I think the data presented in this study are of good quality, especially considering the limitations of a non-model invertebrate animal. The finding that Par-6 may polarize in early embryonic cells but not in later cells undergoing gastrulation is particularly interesting. Unfortunately, the manuscript as presented contains numerous statements and conclusions that are far too strong given the data presented. The authors seem determined to draw strong, sweeping conclusions about metazoan evolution, but these claims rest largely on negative results. A comprehensive study of epithelial organization and polarity in ctenophores is clearly of great importance, but this has not been achieved here.

I particularly took exception to the authors' statement that "Despite of [sic] the high structural conservation of Par proteins across Metazoa, we have shown that ctenophore cells do not deploy and/or stabilize the asymmetrical localization of these proteins" (subsection “Evolution of cell polarity and epithelial structure in Metazoa”). In fact, the authors have examined two out of the dozen or so key conserved proteins that participate in Par polarity in metazoans. One of these (Par-6) localized in a polarized fashion in some, though not all, of the stages examined. The other (Par-1) was difficult to visualize by immunostaining, perhaps in part due to poor performance of the antibody, but it appears to show the opposite polarity from Par-6 at some stages as would be expected (Figure 3D-E). Thus, the data not only do not support the conclusion that "ctenophore cells do not deploy and/or stabilize the asymmetrical localization of these proteins;" they actually suggest the opposite (albeit not conclusively). The data also do not rule out roles for Crumbs, Scribble, Lgl or other conserved Par proteins in polarizing ctenophore cells, nor do they exclude the possibility of Par-dependent polarity at adult stages not examined here. I would urge the authors to be cautious and focus on what the data show, rather than attempting to interpret broadly from a limited number of observations.

- The authors assert that ctenophore Par-3/Par-6/aPKC and Par-1 "contain all the domains present in other metazoans" (subsection “Par protein asymmetry is established early but not maintained during *M. leidyi* embryogenesis”), but no evidence is presented to support this statement.

- The statement that "The components of the ctenophore MPar/aPKC complex… are not only able to phosphorylate and displace MPar-1 and MLgl to the cytoplasm but are also able to interact with Crb and localize to the apical cortex" (subsection “Par protein asymmetry is established early but not maintained during *M. leidyi* embryogenesis”) should be removed from the manuscript. This statement is far too strong given the data presented, which do not in any way examine the interactions between aPKC and Par-1/Lgl or Crb nor provide any evidence of active phosphorylation by aPKC or Par-1.

- The statement "This suggests that none of the lateral polarity proteins (Dlg, Lgl, and Par-1) can localize the lateral cortex of the ctenophore cells" (subsection “Par protein asymmetry is established early but not maintained during *M. leidyi* embryogenesis”) is also inappropriate and should be removed, as only Par-1 has been examined here.

- I object to the authors' definition of a "true-epithelium" as being dependent on Par proteins. Although this point is somewhat semantic, I think that the most useful definitions of epithelia are based on their structural and functional properties. Molecular pathways that remain incompletely understood are a poor basis for these kinds of definitions, since it makes the definitions prone to change as our understanding of molecular mechanisms evolves. Better to simply separate epithelia into "Par-dependent" and "Par-independent" categories and avoid unscientific arguments about what constitutes a "true" epithelium.

- It is not obvious from the images presented in Figure 4—figure supplement 2 that "*Mnemiopsis* clearly have polarized epithelia" (subsection “Evolution of cell polarity and epithelial structure in Metazoa”). The authors show some peripheral actin staining, but the cells themselves look rounded and non-polarized, and not obviously epithelial like.

Authors' response to previous reviews:

To avoid bias, I avoided reading the previous decision letter and the authors' response until after I had written the review above. Having now examined the previous comments, I believe the authors' response is satisfactory. The issues raised by the previous reviewers were, in my opinion, far less significant and serious than the issues of overinterpretation that most struck me upon a naïve reading of the manuscript.

Of note, in their response to the previous reviewers, the authors point out that the mRNA injection experiments and live imaging are very technically challenging, making it difficult to visualize Par proteins in live embryos (responses to point 5). The senior author of this paper is known in the community as an exceptionally skilled microinjector, so if he describes the injections as "really challenging," that statement should be taken seriously. I raise this point because I think the authors should point out the technical difficulty of the mRNA injection experiments in the main manuscript text. I get the impression that the authors have higher confidence in their immunostaining data than in the localization of ectopically expressed proteins; if this is true, then it needs to be conveyed to the reader.

---

## [Author Response]

Summary:This paper by Salinas-Saavedra and Martindale reports on a molecular characterization of epithelial organization in ctenophores using immunohistochemistry and expression of fusion mRNAs. The paper focuses on the endogenous, polarity-associated Par proteins Par-1 and Par-6 and comparing these to fusion mRNAs at the same developmental stages. The authors heterologously express ctenophore Par fusion mRNAs in the cnidarian, Nematostella vectensis. They show that while in Nematostella, ctenophore Par proteins localize in a similar fashion to endogenous cnidarian Par proteins, but that in the ctenophore, based on their localization patterns during embryonic development, they conclude that Par-6 localizes asymmetrically to the apical surface of blastomeres early in development but then later fails to localize in epithelia. They also show that ctenophore Par-1 is primarily localized to cytosolic puncta during most stages of development.Essential revisions:While the results are certainly intriguing, and we commend the authors for generating key reagents that will be critical in addressing the evolutionary origin of metazoan epithelium there are several major concerns that would need to be addressed before the work is suitable for publication. We list these below:1) A more rigorous demonstration that the cell cycle is not affecting protein distribution is necessary. For example, evidence that mitotic cells may have different localization of Par-6 than during interphase is shown in the supplemental figure of MlPar-6 Ab staining showing 4 hpf Animal view. Here, Par-6 localization appears in two cells that are likely in anaphase and there is Par-6 localization polarized to opposite sides of the daughter cells.

We have added a new supplemental figure (Figure 2—figure supplement 8) and its data source file (Figure 2—figure supplement 8—source data 1), where we included intensity measurements along the apical cortex at different phases of the cell cycle (observed throughout RAW images). These new data saw no changes in protein localization, neither in the cortex nor in the cell-cell contacts of blastomeres and ectodermal cells throughout the cell cycle in cells of 125 embryos examined (total number from Figure 2—figure supplement 7 and Figure 2—figure supplement 8).

The differences noted by the reviewers correspond to the specific behaviour of individual cells. As we mention in the text, Par6 only localizes in ‘static’ ectodermal cells. The cells pointed out by the reviewers are undergoing asymmetrical cell divisions and are located into the inner cell layers. Ctenophore cells are attached by persistent mid-bodies that can give the impression of the presence of a mitotic spindle, but it is not the case and these cells are in interphase.

2) Given the nature of the claims made in the paper, it is necessary to quantify the imaging data reported. Making a generalization of localization based on single representative images without quantification is not acceptable. While the authors claim in the body of the text (subsection “MlPar-6 gets localized to the apical cortex of cells during early M. leidyi development”) to have looked at hundreds of embryos, it would make more sense to quantify the localization using FIJI/ImageJ at each developmental stage and then display the representative image that best shows the quantified data as well as include how many embryos were examined at each developmental stage.

We have added a new supplemental figure (Figure 2—figure supplement 7) and its source data file (Figure 2—figure supplement 7—source data 1), where we include these measurements. In the respective source data we have included the raw and normalized data for these measurements. For the earliest stages, we could only do this for embryos where the animal-vegetal axis was perpendicular to the z-axis of the z-stack. For later stages, we took measurements of two cells located in perpendicular axes of the embryo where the apicobasal axis was clearly detectable.

Without being able to examine all of the collected image data, it seems problematic to make a generalization of localization based on single representative images without quantification. For example, for the early blastomere localization of Par-6, it would be important to show the fluorescence intensity of the apical vs. basal side of the population of blastomeres to make a claim about the establishment of polarity, especially during the earliest cell divisions. By performing a quick line scan in FIJI of the data across the cells, it is not particularly convincing that the signal is polarized to the apical cortex.

We performed the intensity measurements using the RAW confocal images without contrast modification as is presented in the Figures. These measurements (Figure 2—figure supplement 7) support the higher intensity observed in the animal/apical cortex. Ctenophore cells are polarized with the apical cortex facing the external media. Thus, we measure cells at the middle point of the z-stack where the apico-basal axis is clearly differentiated.

At first cleavage, there is just as much signal in the lateral right portion of the embryo as there is at the cytokinetic furrow.

We made the respective measures along the animal-vegetal axis. The increased intensity in the right side of the embryo is an artefact of the mounting. The early ctenophore cleavage program is highly symmetrical, and in this preparation, the left side where the morphology is intact, has lower intensity than the animal cortex. Unfortunately, mounting early stages of ctenophores embryos is challenging due to the amount of yolk, density, and delicate consistency. The embryo presented in Figure 2, was the best preparation for this stage. We apologise for this inconvenience, but at the moment, this is beyond our technical skills.

3) It would also be helpful to include a maximum projection of the collected Z-stack of each representative image to highlight the localization of the proteins.

Figures presented in the main article correspond to projections of a portion of the z-stack, unless indicated otherwise. The problem with including maximum projections of the entire embryo is that makes it even more difficult to identify subcellular localization of proteins in a ‘thick’ tissue. We have tried to give the reader an accurate representation of the localization, but realize there is probably not one ‘ideal’ way to convey all of these data in a single snapshot. Please, see Author response image 1. We are happy to provide these images if is necessary and informative to the reader.

4) It is unclear how the Ragkousi et al., 2017 is considered in the schematic. Ragkousi et al., show that Nematostella embryos lose polarity and the localization of Par-6 only during mitotic rounding, yet re-establish polarity during each interphase when they reinforce cell-cell contacts and are compacted.

We have now added the Par proteins in the cell-cell contacts in our schematics. As is observable in the figures of Ragkousi et al., paper, and similar to what we reported two years before Ragkouski et al., (see Salinas-Saavedra et al., 2015), Par proteins (Par6) are not asymmetrically localized in the blastomere cortex but are concentrated in blastomere-blastomere contacts during the earliest cleavage stages.

We tried to make the schematics as simple as possible and considered the most common characteristics of *Nematostella* cell polarity. We are only using as a reference the earliest stages and post gastrula stages of *Nematostella* development to compare with other animals.

The Ragkousi et al., paper reports the events of stereotyped and synchronous cell divisions in *Nematostella* that are rare among the other patterns of cell divisions (Fritzenwanker et al., 2007; Salinas-Saavedra et al., 2015).

As the authors have primarily examined static time points, it is important to take into account cell cycle state, particularly the difference between interphase and mitotic cells. For example, in many of the images shown in the figures the cells are undergoing mitosis. It might be important to compare localization between interphase cells.One striking example where mitotic cells may have different localization of Par-6 than during interphase is the image from the supplement showing Par-6 localization where to the left and right of center there are two cells that appear to be in anaphase and there is Par-6 localization polarized to opposite sides of the daughter cells (supplemental figure of MlPar-6 Ab staining showing 4 hpf Animal view).

Ctenophore cells are connected by their mid bodies, which should not be confused with the mitotic spindle. As can be seen in the Figures, most of the cells are in interphase, specially at 4hpf.

Data from Nematostella (Ragkousi et al., 2017) demonstrates that localization changes of Par proteins between interphase and mitosis during early cleavages of blastomere (more on the interpretation of this data below). It would be extremely interesting to know if something similar is occurring during early Mnemiopsis cell divisions.

This seems not to be the case in ctenophore cells. We have now included intensity measurements along the apical cortex during different phases of the cell cycle with no apparent changes. We show anaphase and telophase data in the new Figure 2—figure supplement 8 of several stages. Intensity measurements reported in Figure 2—figure supplement 7 are of cells going though interphase and metaphase.

5) In reference to figure 2—figure supplement 4B which shows the localization of MIPar-6-mVenus – it is hard to see how similar this localization pattern is to the endogenous Par6.

It is rather challenging to do microinjections in ctenophores, and even more, to overexpress the protein for in vivo imaging. The cell cycle is quite short (15 minutes) and low mRNA concentrations are needed such that it takes several hours before enough protein is expressed to visualize. This means that we are reporting a mixture between the endogenous (unlabelled) and exogenous (labelled) protein and we were only able to observe where the protein is highly concentrated (hence, we add this data as a supplement). Antibody staining, on the other hand, shows the total protein in the cells. These differences are expected due to the difficulty in the handling and imaging. However, both set of experiments show similar localization enriched at the cell-cell contacts for later stages. Ctenophore embryonic material is ‘exquisite.’ However, it presents a number of technical limitations (perhaps that is why there is not a large literature on them).

For example, the antibody staining for Par6 at 5hpf shows an enrichment in the cells on the animal pole, but this localization is not shown with the MIPar-6-mVenus expressing animals.

In vivo imaging of ctenophore embryos is a real challenge. The signal, as every in vivo imaging, is faint due to the thickness of the tissue and lack of tissue clearing. In addition, mounting specimens in the ideal position for imaging is problematic. For this specific stage, we were not able to find the right position to image the whole embryo and make a figure from that. We think that the immune histochemical analyses are more thorough. Unfortunately, as we explain below, these experiments cannot be repeated in a near future.

The authors should fix the MIPar-6-mVenus expressing embryos and stain for Par6.

We tried this. However, ctenophore embryos disintegrate when they are fixed without the chorion (needed for microinjection) and the addition of even a small amount (over 0.2%) of glutaraldehyde (which helps stabilize cellular structure) causes autofluorescent background. In addition, microinjections are really challenging and we were able to inject and overexpress the protein into a few number of embryos.

Unfortunately, these experiments cannot be accomplished in a near future for two main reasons: (1) MSS, is now a postdoc in Ireland, and (2) Coronavirus pandemic and lockdowns prevent travel back to the Whitney Lab where these experiments were performed. These expression studies were performed to validate the immunohistochemistry localizations, and with the hope of additional live cell imaging that did not come as easily as we had hoped.

6) In reference to the summary model – it is unclear how interpretation of the Ragkousi et al., paper is considered in the schematic. In their paper, they showed that Nematostella embryos lose polarity and the localization of Par-6 only during mitotic rounding but re-establish polarity during each interphase when they reinforce cell-cell contacts and are compacted. It would be extremely interesting to know if something similar is occurring during Mnemiopsis cell divisions early.

See response above in point 4.

Also, it would be worth more carefully visualizing the heterologous experiments or including images of compacted Nematostella embryos expressing ctenophore Par-6 to verify that it localizes in the same manner as the endogenous NvPar-6.

These expression studies were performed to validate the structure conservation. In that moment, we did not carefully look for changes in cell cycle. Unfortunately, we are unable to do these experiments by the reasons given in point 5 (inability of travel).

[Editors' note: further revisions were suggested prior to acceptance, as described below.]

We thank the authors for preparing this revised version of their work showing that Par6 polarizes in early but not late blastomeres. The manuscript was re-evaluated by one of the original reviewers (reviewer #2) and a new reviewer (reviewer #3), who prepared their independent thoughts and then evaluated the prior reviews and the authors' response to reviewer. As you will see below, reviewer #2 still has some remaining concerns, particularly about statistical analysis and the need for a schematic, whereas reviewer #3 has more extensive concerns about over-interpretation in the manuscript. In the post review discussion, reviewer #2 agreed that reviewer #3's concerns are valid and that they would also like to see them addressed. I am therefore inviting a further revised version (which is rarely done at eLife) so that you can address these issues. The reviewers agree that no new data is required. Specifically, they think that their "comments could be addressed by text revisions, especially toning down the overly strong conclusions. The authors also need to show some sequence alignments, and ideally also domain prediction, to support their claims about the conservation of Ctenophore PAR proteins; but this would only require computational work rather than new experiments."

We thank the reviewers and their comments. Information about ctenophore biology is not abundant and we appreciate all the help given by the reviewers to analyze our data. In this version we have changed the tone of the main text and addressed the changes suggested by the reviewers. We have also included schematics and statistical data of our intensity measurements and better explanations for the specific cases pointed out by the reviewers. Finally, we have added a new protein alignment for Par6 and Par1 with their respective protein domains. Please, find below our detailed response for the full comments.

Reviewer #2:We commend the authors for responding to our previous comments. We also thank the authors for clarifying that ctenophore interphase cells maintain mid-bodies and apologize for a lack of familiarity on these fascinating embryos. In regards to the addition of the requested quantification of their data, we have several concerns (specifics detailed below). Anytime quantification is done, it is appropriate to also include statistical tests to support claims (of either no change in localization across an area measured or a significant change across the area measured). Additionally, it would be very helpful to the readers to include a schematic of where the line scans were done – either utilizing the staged drawings of the ctenophore embryos or placing a line on an example micrograph.

Our apologies. We agree that these additions will help clarify our results. We now included these data and schematics in Figure 2—figure supplement 7, Figure 2—figure supplement 8, Figure 2—figure supplement 9 and Figure 2—figure supplement 10 and Figure 3—figure supplement 5 and Figure 3—figure supplement 6.

Specifically:Figure 2—figure supplement 7 – Line scans appear very noisy for 1 and 2 cell stage measurements (i.e., there is a large degree of variability between the 4 quantified embryos at the 1-cell stage and 1/3 embryos at the 2-cell stage is quite different than the other 2)

Thank you for pointing this out. We have now explained the situation with these embryos in the figure legend of Figure 2—figure supplement 7. The early stages of ctenophore development are quite delicate (before many blastomere-blastomere cell contacts form). For this particular case, the measurements were taken on a preparation where part of the cytoplasm was not in the same focal plane as the cortex and therefore presented this lack of intensity (see Author response image 2). Nevertheless, the blastomeres are still intact and do not present much variation, as can be seen in the maximum projections presented below this answer. As we explained in our previous revision, making the measurements in maximum projections is not appropriate because of the noise added by this mounting artefact.

**Author response image 2. respfig2:** 

– at the 4 cell stage there seems to be a clear enrichment on apical vs basal, but this would require statistical tests to make the appropriate claim. – or relative enrichment on apical vs basal would make this quantitatively clear and allow the authors to make (or not) their claims.

Added in Figure 2—figure supplement 8, Figure 2—figure supplement 8, Figure 2—figure supplement 9 and Figure 2—figure supplement 10 that quantifies the differences described.

Figure 2—figure Supplement 8Here, the argument for telophase enrichment only has an n=2 embryos for both 2 and 4 cell stages – in which one embryo of the 2 does seem to have a dramatic changes in enrichment along the apical surface. We recognize the difficulties associated with imaging these embryos but the data needs to be consistent across samples to make claims, even where statistics may not be able to be used due to low sample size (but some sort of claim needs to be made stating that a statistical analysis could not be performed due to this low (n) issue).

We have added a statement in the figure legend of Figure 2—figure supplement 11: Unfortunately, we did not have enough telophase replicates to show statistical significance of these observations. However, we think the dramatic changes correspond to the larger size of the cells and the cleavage furrow which are explained in Figure 2—figure supplement 7 schematics.

Additionally, we had requested that the authors examine apical vs. basal enrichment during the cell cycle, but it appears that they have quantified the distribution of signal across the apical cortex only,

The quantification to examine apical vs. basal enrichment shown in Figure 2—figure supplements 7, Figure 2—figure supplement 8, Figure 2—figure supplement 9 and Figure 2—figure supplement 10 also represent cycling cells and we show that the apical localization of Par-6 seems unaffected by cell cycle dynamics. This can be appreciated in Figure 2Db’ where the arrow points to a cell in anaphase. In Figure 2—figure supplement 7 we have also included a cell in metaphase. Hence, we think that including measurements along the apical cortex is also valuable because it shows the signal distribution in the cleavage furrow.

but for example at first cleavage there is a dip in signal at the mid-point of their line scan, but in the text they make the conclusion that there is more Par6 at the cleavage furrow – or is the dip in signal overlapping with the absence of localization right at the middle of the cleavage furrow (Figure 2B/A'). Again, a schematic or annotation of how the line scans were made would be extremely useful for interpretation of data.

The reviewer is correct. The dip in the signal is due to the empty space created by the cleavage furrow. We have clarified this in the Figure 2—figure supplement 7 with schematics.

Figure 3 and Figure 4 – Really, to make any claims about localization, the quantification should be included in the main figures as well as an annotation of where the line scan was made in a representative image along with reporting the (n) analyzed and the appropriate statistical tests included and referenced in the main text manuscript.

We apologize for the lack of schematics in the first revision. Now we have added new figures where the statistics are graphically represented (Figure 2—figure supplement 9, Figure 2—figure supplement 10 and Figure 3—figure supplement 5) and we have also added new schematic figures to show how all these measurements were made to Figure 2—figure supplement 7 and Figure 3—figure supplement 6.

We are submitting this work as a Short Report and we would like to limit the number of main figures to show the immunostaining of these proteins in the largest sample of developmental stages as possible. We think that these figures are already large and saturated with information. For aesthetic reason and simplicity, we would like to keep the quantification as supplemental figures for Figure 3. We have added this data in Figure 4. We think this should not be a problem given the *eLife* online format. If the reviewers think that this is a critical problem, we are willing to rearrange our figures.

Reviewer #3:It is widely accepted that polarized epithelial tissues are a conserved feature of all metazoans, although the vast majority of studies have focused on bilaterian model organisms. Here, Salinas-Saavedra and Martindale examine cell polarity in M. leidyi, a representative of the basally branching ctenophore lineage. Using immunostaining and mRNA injections, the authors find that two well-studied polarity proteins, Par-6 and Par-1, exhibit a polarized localization in a smaller number of cells and during a narrower range of times during M. leidyi embryogenesis than might have been predicted based on studies in other systems.The examination of epithelial polarity in an early-branching metazoan lineage is interesting and important, and I think the data presented in this study are of good quality, especially considering the limitations of a non-model invertebrate animal. The finding that Par-6 may polarize in early embryonic cells but not in later cells undergoing gastrulation is particularly interesting. Unfortunately, the manuscript as presented contains numerous statements and conclusions that are far too strong given the data presented. The authors seem determined to draw strong, sweeping conclusions about metazoan evolution, but these claims rest largely on negative results. A comprehensive study of epithelial organization and polarity in ctenophores is clearly of great importance, but this has not been achieved here.

We acknowledge that our observations are only the beginning of a comprehensive understanding of the structure and function of ctenophore epithelial cells and the molecular basis of cell polarity. However, what we have seen in *Mnemiopsis* is rather radically different than what we have seen in *Nematostella* (which in itself was different from other bilaterians, at least during early cleavage stages). Because the reagents and techniques we deployed in *Nematostella* are pretty much the same we used in *Mnemiopsis* we feel that the differences are in fact biological. We probably over interpreted the meaning of these findings in our original versions but since these are the very first observations in this system we have few data by which to compare our results to.

I particularly took exception to the authors' statement that "Despite of [sic] the high structural conservation of Par proteins across Metazoa, we have shown that ctenophore cells do not deploy and/or stabilize the asymmetrical localization of these proteins" (subsection “Evolution of cell polarity and epithelial structure in Metazoa”). In fact, the authors have examined two out of the dozen or so key conserved proteins that participate in Par polarity in metazoans.

The reviewer is correct. We have only shown two proteins of the Par complex and now we have added two new supplementary figures (Figure 1—figure supplement 2 and Figure 1—figure supplement 3) to show sequence alignments to other metazoan species. We have also changed this statement to be more specific: ‘*ctenophore cells do not deploy and/or stabilize the asymmetrical localization of Par-6 and Par-1 proteins’*

We have cloned *Mnemiopsis* aPKC and CDC42 and tried to see their in vivo localization by mRNA microinjection as well. However, the expression of these constructs was not strong enough to have conclusive information and we did not observe any distinctive localization. As we stated in the text, we were not able to clone other proteins to test in vivo.

Regarding the conservation of the full set of proteins, we have made the respective analyses within the genomic sequences and transcriptomics when we started to work with these proteins back in 2016. We decided to do not report these studies because Belahbib et al., 2018 (referenced) have done a more complete analysis for most of these proteins and they deserve full credit for that.

One of these (Par-6) localized in a polarized fashion in some, though not all, of the stages examined. The other (Par-1) was difficult to visualize by immunostaining, perhaps in part due to poor performance of the antibody,

We performed the appropriate experiments to test this antibody and we are confident of its quality. In addition, we rely in the in vivo protein expression as an extra confirmation.

but it appears to show the opposite polarity from Par-6 at some stages as would be expected (Figure 3D-E). Thus, the data not only do not support the conclusion that "ctenophore cells do not deploy and/or stabilize the asymmetrical localization of these proteins;" they actually suggest the opposite (albeit not conclusively).

We are including the respective quantification of the fluorescence to show that these apparent differences are not significant (Figure 3—figure supplement 5.).

The data also do not rule out roles for Crumbs, Scribble,

Scribble and proteins of the crumbs complex are not present in ctenophore genomes (Belahbib et al., 2018).

Lgl or other conserved Par proteins in polarizing ctenophore cells, nor do they exclude the possibility of Par-dependent polarity at adult stages not examined here. I would urge the authors to be cautious and focus on what the data show, rather than attempting to interpret broadly from a limited number of observations.

The conservation of these proteins has been published already (Belahbib et al., 2018) and the two ctenophore genes we studied here polarize when expressed in *Nematostella*. We assumed that the lack of a Scribble complex in ctenophores may not be able to localize Lgl, but the reviewer is correct. We do not have a mechanistic answer for this at this time.

Studying fully grown adult ctenophores is difficult, but since ctenophores are ‘direct developers’ and their adult body plan is generated rapidly (24 hours.) we feel relatively confident that epithelial organization is similar during older adult stages.

- The authors assert that ctenophore Par-3/Par-6/aPKC and Par-1 "contain all the domains present in other metazoans" (subsection “Par protein asymmetry is established early but not maintained during M. leidyi embryogenesis”), but no evidence is presented to support this statement.

These analyses have been published already (Belahbib et al., 2018). We have now included the protein alignment from the sequences cloned in our study (Figure 1—figure supplement 1, Figure 1—figure supplement 2 and Figure 1—figure supplement 3).

- The statement that "The components of the ctenophore MPar/aPKC complex… are not only able to phosphorylate and displace MPar-1 and MLgl to the cytoplasm but are also able to interact with Crb and localize to the apical cortex" (subsection “Par protein asymmetry is established early but not maintained during M. leidyi embryogenesis”) should be removed from the manuscript. This statement is far too strong given the data presented, which do not in any way examine the interactions between aPKC and Par-1/Lgl or Crb nor provide any evidence of active phosphorylation by aPKC or Par-1.- The statement "This suggests that none of the lateral polarity proteins (Dlg, Lgl, and Par-1) can localize the lateral cortex of the ctenophore cells" (subsection “Par protein asymmetry is established early but not maintained during M. leidyi embryogenesis”) is also inappropriate and should be removed, as only Par-1 has been examined here.

They were only suggestions and now have been removed.

- I object to the authors' definition of a "true-epithelium" as being dependent on Par proteins. Although this point is somewhat semantic, I think that the most useful definitions of epithelia are based on their structural and functional properties. Molecular pathways that remain incompletely understood are a poor basis for these kinds of definitions, since it makes the definitions prone to change as our understanding of molecular mechanisms evolves. Better to simply separate epithelia into "Par-dependent" and "Par-independent" categories and avoid unscientific arguments about what constitutes a "true" epithelium.

This is a fair point and is mostly semantic, so we have changed it as the reviewer suggested. However, we clearly defined what we meant by a ‘true’ epithelium (which of course one could always argue with!). Our definition aimed to convey structural and functional properties of bilaterians and cnidarians. Ctenophora do not have many of those structural components, e.g. Wnt PCP pathway, Crb complex, Scribble, classical e-cadherin, SJs, and our results show that Par proteins do not localize as we had expected. Hence, we decided to use a mechanistic definition. Now we have changed to a "Par-dependent" and "Par-independent." Discussion section.

- It is not obvious from the images presented in Figure 4 —figure supplement 2 that "Mnemiopsis clearly have polarized epithelia" (subsection “Evolution of cell polarity and epithelial structure in Metazoa”). The authors show some peripheral actin staining, but the cells themselves look rounded and non-polarized, and not obviously epithelial like.

Thanks for pointing this out. We still believe that *M. leidyi* has a polarized epithelium given the presence of a cilium in the apical side of epithelial cells. This can be observed in the figures after 20 hpf. However, we do not have an elegant picture to show the intact epithelium with all its characteristics at this point. To avoid confusion and unnecessary statements, we have deleted this figure and the correspondent text.

Authors' response to previous reviews:To avoid bias, I avoided reading the previous decision letter and the authors' response until after I had written the review above. Having now examined the previous comments, I believe the authors' response is satisfactory. The issues raised by the previous reviewers were, in my opinion, far less significant and serious than the issues of overinterpretation that most struck me upon a naïve reading of the manuscript.Of note, in their response to the previous reviewers, the authors point out that the mRNA injection experiments and live imaging are very technically challenging, making it difficult to visualize Par proteins in live embryos (responses to point 5). The senior author of this paper is known in the community as an exceptionally skilled microinjector, so if he describes the injections as "really challenging," that statement should be taken seriously. I raise this point because I think the authors should point out the technical difficulty of the mRNA injection experiments in the main manuscript text. I get the impression that the authors have higher confidence in their immunostaining data than in the localization of ectopically expressed proteins; if this is true, then it needs to be conveyed to the reader.

We thought it was important to validate our immunohistochemical studies. Now we have stated this: ‘Microinjection and mRNA expression in ctenophores is really challenging. For the first time, we have overexpressed fluorescent-tagged proteins for in vivo imaging. In spite of the low number of replicates (see Materials and methods section), our results are consistent with the antibody observations presented above.’